# Patterns of Injury in the Spanish Football League Players

**DOI:** 10.3390/ijerph19010252

**Published:** 2021-12-27

**Authors:** Iván Prieto-Lage, Juan Carlos Argibay-González, Adrián Paramés-González, Alexandra Pichel-Represas, Diego Bermúdez-Fernández, Alfonso Gutiérrez-Santiago

**Affiliations:** Observational Research Group, Faculty of Education and Sport, University of Vigo, 36005 Pontevedra, Spain; ivanprieto@uvigo.es (I.P.-L.); aparames@uvigo.es (A.P.-G.); alexandrapichel19@gmail.com (A.P.-R.); diegobf87@gmail.com (D.B.-F.); ags@uvigo.es (A.G.-S.)

**Keywords:** injury, football, pattern, video analysis

## Abstract

Background: The study of football injuries is a subject that concerns the scientific community. The problem of most of the available research is that it is mainly descriptive. The objective of this study is to discover and analyse the patterns of injury in the Spanish Football League (2016–2017 season). Methods: The sample data consisted of 136 given injuries identified by the official physicians of the football clubs. The analysis was performed by using traditional statistic tests, T-pattern detection and polar coordinate analysis. Results: The analysis revealed several patterns of injury: (a) The defender suffered a rupture of the hamstring muscles after a sprint, (b) knee sprains happened due to a received tackle, (c) fibrillar adductor rupture appeared mostly among defenders and (d) fibrillar ruptures took place mostly throughout the first part. Conclusions: There is a marked shift in the tendency regarding the player who gets more injured, from the midfielder to the defender. The most common injury was fibrillar rupture. The most common scenario in which this injury occurred was that in which the player injured himself after a sprint (24%). A week without competing seems to be insufficient as a prevention mechanism for injuries.

## 1. Introduction

The Spanish Football League (LaLiga) is one of the main sports competitions in the world, with a high economic and social impact, representing 1.37% of the Spanish GDP, generating around 200,000 jobs per year and generating an amount of more than 4000 million euros per year in tax contributions [1]. Studies on similar competitions to the Spanish one have calculated that an injury can cost up to 500,000 euros if the injury period is 1 month [2]. Likewise, an investigation of the English Premier League teams determined that in the 2013–2014 season, for injuries of more than 30 days, clubs would pay more than £100,000,000 in wages [3].

Throughout the past decade, several studies have been carried out on injures in football [4]. The number of injuries is higher in a competition than in training due to the number of minutes of exposure [5,6]. Authors have shown that 25–28 injuries are suffered for every 1000 h of competition and 7–10 injuries occur for every 1000 h of training [4,5,6,7,8,9]. Additionally, the largest percentage of injuries happen in the lower extremities (70–95% compared to other parts of the body) [5,6,8,9,10]. If we take into account the leg that suffers the injury, some authors [11] indicate that the dominant leg suffers the injury in 55.8% of the cases, while the non-dominant leg suffers 34.3% of the injuries. Similar results have been found in previous studies [12,13].

The most frequent injuries are in the thigh (especially the back section), knee, ankle, and groin [7,13]. As found lately [5], 1% of the injuries are caused by hard tackles and, therefore, punished by the rulebook. Among that percentage, the most common ones are ankle sprain, at 15%, while knee sprains were only 9% [7,13]. The studies differ among the way the injury is suffered; some indicate that up to 80% of the injuries take place through contact [14], while others state the opposite (just 20%) [12]. Other research provides more even data, at 46–54% [15] and 42–58% [13].

In contrast, some studies [4,9] point out that the nature of the injury is not the same in a competition and in training, because traumatic injuries happen more often in a competition, while the injuries suffered in training are often more related to overuse.

The midfielder is the position that suffers more injuries (37.6%), followed by the defenders (29.6%), strikers (20.5%) and, finally, the goalkeepers (8.3%) [8].

There are no significant differences in the distribution of injuries comparing the first half with the second half, although there are significant differences in the distribution in the way that injuries take place, because injuries without contact are common during the second half [9]. Moreover, the occurrence of injuries increases significantly as the time in each half advances, as demonstrated in previous studies [5,13,16].

Research has revealed that there are no significant differences between different age groups (under 25 years old, between 25 and 30 years old, and older than 30 years) while studying the seriousness and rate of the injuries [8].

Taking into account how advanced the season is, data indicate that higher injury occurrence takes place during the preseason [9]. If we take into account the training, most of the injuries occur during the first part of the season; but if we only concentrate on the competitions, the opposite occurs—the higher injury occurrence happens along the second half of the season.

It goes without discussion that the study of football injuries is a subject that concerns the scientific community. The problem of most of the studies is that they are especially descriptive, without establishing what causes these injuries. Therefore, the objective of this study is to discover and analyse the patterns of injury in the Spanish Football League during the 2016–2017 season. The results of this research will generate knowledge of the patterns of injury of football players, which will be useful to coaches, trainers and players to improve the training and competition systems.

## 2. Materials and Methods

### 2.1. Design

We used the observational method because it allowed us to study the player’s actions that lead to an injury in a natural way and with the necessary rigor and flexibility. The kind of observation carried out was systematic, open and non-participant [17].

The observational design [18] used was nomothetic (all injuries were studied independently), follow-up (one season) and unidimensional (one level of response). A series of decisions about the participants, instruments and the analytical process was derived from this design.

### 2.2. Participants

The participants of this study were the players of the first division of the Spanish Football League for men in the 2016–2017 season who suffered an injury because of which they were withdrawn from the field. A sample of 136 injuries was obtained from the 38 league days. The information about the injuries was taken from the official medical records of the clubs and from TransferMarkt, a valid sports database [19] that provides information about injuries. The injuries were analysed in accordance with the ethical principles of the Helsinki Declaration using audiovisual material in the public domain [20]. According to the American Psychological Association [21], an observational study in a natural environment, with public videos obtained from the mass media that does not imply experimentation, does not require informed consent from the participants. The study was approved by the ethics committee of the Faculty of Education and Sport Science (University of Vigo, application 02/1019).

### 2.3. Instruments

The observation instrument designed ad hoc for this study is OI-INJURIES-FOOTBALL, a category system that contemplates a collectiveness of criteria that allows us to determine the football injuries’ characteristics [18]. Each dimension gives rise to a system of categories that accomplish the conditions of exhaustiveness and mutual exclusivity (E/ME). A detailed description of the observation instrument appears in Table 1, where the criteria, categories and subcategories of the instrument are shown. In Figure 1, the established field areas are detailed.

The categorisation of the type of injury followed the UEFA model [22]. To designate the part of the body where the injury occurred, a classification endorsed by other authors was used [23,24,25]. Four demarcations were used for the analysis [14,19,25,26] and divided the match into 15 min intervals [5,9].

All the injuries were codified using the software LINCE v.1.4. [27]. With this instrument, all the data were registered. This software is a free multimedia interactive programme that allows simultaneous viewing and registering of the filmed material in a computer that is used to support the observational analysis in a systematic way. This software has been used in numerous investigations of football [28,29].

### 2.4. Procedure

The sample was obtained through the Wyscout platform [30], a paid online scouting platform for football.

Behind the design of the observational instrument OI-INJURIES-FOOTBALL, the validity of its construct was determined through its coherence with the theoretical framework and through a consultation with two experts in the observational methodology, injuries and football, that needed to show the degree of agreement, achieving a level of agreement of 92%.

After adequate training in the use of the register instrument and the observational instrument OI-INJURIES-FOOTBALL, the data were registered by two observational experts. To guarantee rigor in the codification process [31], the quality of the registered data was controlled through a calculation of the intra- and inter-observers’ compliance using Cohen’s kappa coefficient calculated using the software LINCE. The intra-observer compliance was previously calculated among a third of the injuries (n = 45; not belonging to the final sample), obtaining a kappa value of 0.97 for observer 1 and 0.86 for observer 2. Subsequently, the inter-observer agreement achieved a kappa value of 0.83. Afterwards, the data were recorded by observer 2.

After registering all the injuries, a Microsoft Excel file was obtained with the sequence of all the codes of the registered behaviours, with the temporality and duration expressed in frames. The versatility of this file allowed us to conduct successive transformation for the different analyses: T-patterns and polar coordinates.

### 2.5. Data Analysis

All the statistical analysis was carried out using IBM Statistical Package for the Social Sciences, version 25.0 (IBM-SPSS Inc., Chicago, IL, USA). It calculated the relationship between the different categories that were studied by using the chi-square (χ^2^) test. Statistical significance was assumed as *p* < 0.05.

To analyse the patterns of injury, the detection of T-patterns was carried out using Theme v.5.0 [32], with a significance level of 0.005, which means that the percentage of accepting a critical range due to chance is 0.5%. The minimum number of occurrences in the search of T-patterns was three (the minimum possible without a mistake in processing the information because of an excessive number of series). Furthermore, the reduction in redundancies was activated to 90% to avoid the occurrence of similar T-patterns. This software reveals hidden structures and non-observable aspects in sports science [32,33]. The graphic representation in dendrograms guides the discovery of existing linkages among the different aspects of the injuries. The left quadrant represents the connection among the different categories, which must be read from top to bottom. Meanwhile, the right quadrant allows us to see how many times such connections occur through lines that go from top to bottom.

The sequential analysis of delays was executed through GSEQ5 [34] using only the subsequent calculation of polar coordinates. According to other studies [35], the major delays considered were 1.96, with a significance of *p* < 0.05, that implied a relation of activation between the conduct criteria and the conditioned. The results that were less than or equal to −1.96 were considered equally significant (*p* < 0.05), which was implied in the relation of inhibition between the conduct criteria and the conditioned.

The analysis of polar coordinates was carried out using the HOISAN programme [36] following the analytic technique of Sackett [37] in the variant of genuine retrospective [38], as used in similar studies. We considered as significant (*p* < 0.05) the relations between the focal categories and the conditioned categories when the length of the vector was higher than 1.96. The behavioural connection is determined by the quadrant in which the behaviour is represented and the angle. This way, quadrant I indicates both behaviours are mutually activated in both directions. Quadrant II indicates that the conditioned behaviour activates the focal behaviour, which then inhibits the conditioned behaviour. Quadrant III indicates that the focal behaviour and the conditioned behaviour are both inhibited mutually in both directions. Quadrant IV indicates that the focal behaviour activates the conditioned behaviour, which then inhibits the focal behaviour. The three most common types of injury were used as focal behaviours for the analysis (strain, sprain and contusion).

## 3. Results

### 3.1. Statistical Analysis

In Table 2, a descriptive analysis and the χ^2^ test intra-criteria are presented.

Significant differences were verified among the different categories of the age criterion. Subjects between the age of 26 and 34 years were the ones who suffered injuries more frequently (65%), even though this is the range that refers to the most common age of the studied players. The injured player’s position also showed significant statistical differences. In this case, the defenders (49%) followed by the midfielders (27%) were the most frequently injured players. With regard to the location where the injuries occur more often, it was proven that injuries occur more often in the defensive zone rather than the attacking zone (68% vs. 32%). There were also significant statistical differences (χ^2^ = 75,132; *p* = 0.000) in the most common injury, strains being the most common ones (33%), followed by sprains (24%) and contusions (19%). There were no significant differences if we performed the analysis only with these three types of injuries.

Most of the injuries occurred in the lower extremity (87%) and in the right leg (47%). The months when most of the injuries occurred were September (16%) and April (20%). We did not assess big differences among the injuries suffered in the first and second parts of the season (56–44%) or among the first and second periods of the match (54–46%); nevertheless, we did assess a higher occurrence within the first 15 min of the second half (23%).

There were statistically significant differences between the different ways of getting injured. The players were more often injured alone (52%) after a sprint (24%). Another common way was to be injured by an opponent (48%) after a hard tackle (27%).

Without differentiating the type of injury, these were recorded when players accumulated between 0–500 min in the season (34%). In addition, up to 50% of the injuries occurred before 200 min accumulated after a rest of more than 7 days. There were significant statistical differences in both criteria.

### 3.2. Identification of Temporal Patterns (T-Patterns)

#### 3.2.1. General Description of T-Patterns

The Table 3 below shows an analysis of the number of patterns found through a selective search with three occurrences. As a selection criterion, the presence of the position category was used (defence, midfield and forward) combined with one of three most frequent injuries (strain, sprain or contusion).

The table included below shows the most relevant T-patterns organised by position, type of injury and number of minutes accumulated during the season (total and/or after rest).

#### 3.2.2. T-Patterns in Defenders

Of the 31 strain injuries that were recorded, up to 14 times (45%), the player was injured by a hamstring rupture after a sprint without the presence of an opponent (Figure 2A). The evidence suggests that these injuries occur after less than 200 min accumulated after rest and predominantly in the opponent’s field. Of the 11 adductor rupture injuries observed in total in this investigation, 9 took place in defenders (82%). Strain mostly occurred when the player played between 500 and 1000 min during the season (36%). High values were detected when the player accumulated between 1000 and 1500 min (26%).

Up to 40% of the sprains (4/10) were on the ankle after a jump without the presence of an opposing player and in their own field. The T-patterns also reflected that they occurred indistinctly in the first and second parts of the season, as well as in the first or second half. The predominant age for this injury was between 26 and 34 years.

Most of the contusions affected a lower extremity (8/11, 73%) and were caused by a rival. They happened more frequently in the first part of a match (65%) and in the second part of the season of a competition (55%).

#### 3.2.3. T-Patterns in Midfields

The study did not evidence many strains in midfields (only five). Most of them (60%) were hamstring fibrillar ruptures, in which there was no intervention of an opponent. They were more frequent in the first part of the season (80%) and in the first half of the matches (80%).

Of the 12 sprains registered, 8 (67%) occurred in the first part of the season (before having playing 500 min in the season) due to an opponent’s tackle (Figure 2B).

Contusions were also mostly caused by an opponent (8/9; 89%). They were more frequent in the defensive zone (78%), in the second half (67%) and in the first part of the season (78%).

#### 3.2.4. T-Patterns in Forwards

Most of the strains (6/8) happened in the offensive zone, after a sprint and without the presence of an opponent (75%). They were more frequent in the first part of the season, (63%), in the first half (63%) and with less than 500 accumulated minutes (38%).

Most of the sprains (4/9, 44%), the same as the previously mentioned midfields, were caused by a hard tackle during the first part of the season. Most of them were registered in the first half (67%) after accumulating less than 500 min in the season (56%) (Figure 2C).

Almost all the contusions were caused by an opponent (3/4; 75%) in the second half of the match (75%). The accumulated time played was not relevant.

### 3.3. Analysis of the Polar Coordinates

The results of the analysis of polar coordinates revealed significant statistical connections among the focal behaviours (strain, sprain and contusion) and the conditioned behaviours (the rest of the categories).

The polar coordinate of the strain (Figure 3A) showed that it is more common in defenders and that it occurs in the absence of an opponent (usually after a sprint).

Figure 3B indicates that the sprain was more frequent during the first part of the season and during the first half of matches. Defenders and midfielders were more likely to suffer this type of injury, which occurred after a tackle or a jump.

Contusion (Figure 3C) usually occurred in the second half of the matches and was more common in defenders and midfielders. They were usually caused by a collision or a hard tackle.

## 4. Discussion

It has been confirmed that the most frequent injury is strain, in particular fibrillar rupture (33%), a finding that agrees with the results given in the Bundesliga (30.3%) [19] or with the 31% obtained by other authors [39]. With regard to the injuries of ligaments (sprains), the results found in our study were similar (24%) to those produced in other studies [19]. Analysing the contusions, the values of 19% obtained differ from the ones reported in other studies: 8.5% [19] and 16% [7].

With regard to the way the injury takes place, studies [40,41] agree that most of the injuries occur without previous contact (approximately 70%), a fact that was also proved in our study but at a lower frequency (52%).

As it might seem logical, the lower extremity showed a higher injury percentage (87%), similar to the 89.6% previously reported [9].

With regard to the affected leg, the reported data match those found in this study. Recent studies have shown values of 54.4% (right leg) and 36.5% (left leg) [40], meanwhile others have reported values of 52% and 38.7% [16].

According to the specialised literature, at the end of every half of the match, there is a higher risk of suffering an injury [5,16], a circumstance that is completely opposite to the results of this study, given that the higher occurrence of injury was identified from 15 to 30 min and 46 to 60 min at 20% and 23%, respectively. Considering these last data, it could be sensed that at the beginning of the second period, the players do not manage to get predisposed to the effort at a physiological level.

There are no big differences regarding the number of injuries between the first and the second half (54% and 46%, respectively), something also cross-checked by other studies [4,40], which have revealed 58.1–41.9% and 51–49% injuries between the first and the second half.

As regards the position of the player in the play field, the results are clear: defenders (49%) are the ones who get more injured, followed by midfielders (28%), forwards (18%) and goalkeepers (5%). These results vary from those found in other studies, where the midfielders’ position had the highest percentage: 37.6% [8], 39% [6] and 37.7% [19].

There were differences between the age ranges where an injury occurs in this study; this may be due to the fact that the 26–34-year age range has the most players. This result does not coincide with those found in the Major Soccer League [8], where there were no differences between the three age ranges studied.

The months with the highest number of injuries in our study were September, October and April, similar to what was found in a longitudinal study in the Spanish Football League between 2012 and 2016 [11]. Another investigation in this league in the 2008–2009 season showed that March and May are the months with the highest number of injuries [9].

In contrast to what was reported in the preceding investigation [9], in our case, most of the injuries originated in the first part of the season.

Finally, focusing on the frequency of injuries suffered as a function of total accumulated minutes in the season, and accumulated minutes after rest, it has been observed that the highest number of injuries is found among the categories with the least accumulated playing time (33% in TAM 0–500 and 50% in AMAR 0–200).

The first case can be due to the high amount of training in the preseason, which generates excessive fatigue that translates in an injury at the beginning of the season, something that has been evidenced in other studies [9,13]. For the second case, research that confirms these findings has not been found, although it can be noted that this might occur, in spite of the rest, due to competitive stress or the accumulation of fatigue from the season (weekly rest will not be enough as an injury prevention mechanism).

From a practical point of view, it would be important for coaches to take into account the injury data presented, especially those related to strains, in order to avoid injuries of this type as far as possible. It seems that the players are not physiologically prepared for the effort at the beginning of the second half, so the rest time should not be used exclusively for tactical aspects but also for physical aspects, preparing them for the restart. Weekly rest (>7 days) is not sufficient as an injury prevention mechanism, so coaches should consider rotation systems that provide sufficient rest to prevent injury.

The other types of injuries are usually caused by the opposing player, so it is more difficult to carry out preventive work.

## 5. Conclusions

The most common injury is strain, followed by sprains and contusions. Fibrillar rupture after a sprint and without contact with the opponent is the most frequent injury.

There is a change in the tendency of the player who gets injured the most: before the midfielders and now the defenders.

It has been noted that the rotation of weeks (resting without competing) is not enough as an injury prevention mechanism. The largest number of injuries take place during the first 200 min after the rest period. In addition, players are mostly injured before the first 500 cumulative minutes of the season have elapsed. Therefore, injury prevention programmes during preseason and the start of the season are important.

Defenders are the most injured players, with hamstring strains after a sprint being the most common. In general, fibrillar ruptures occur mainly in the first half. Adductor fibrillar ruptures occur mainly in defenders. The accumulation of minutes in the season in this position (and not in the others) increases the number of strains.

Injuries due to a tackle received or made are also frequent. In this respect, defenders are frequently injured by a tackle made, while midfielders are equally injured by a tackle made or received. Sprains are more common in midfielders and forwards.

## Figures and Tables

**Figure 1 ijerph-19-00252-f001:**
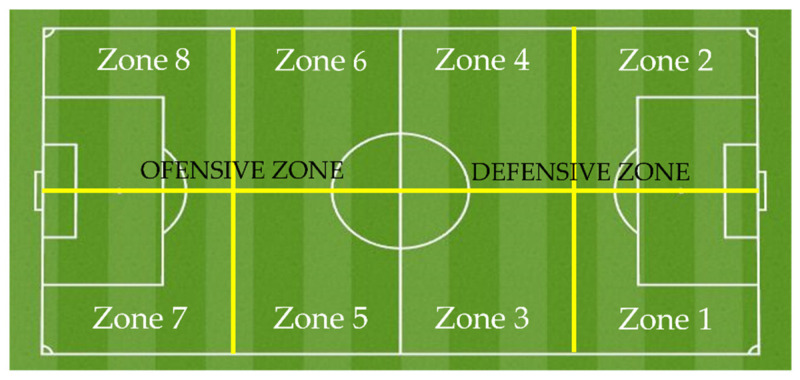
Field areas.

**Figure 2 ijerph-19-00252-f002:**
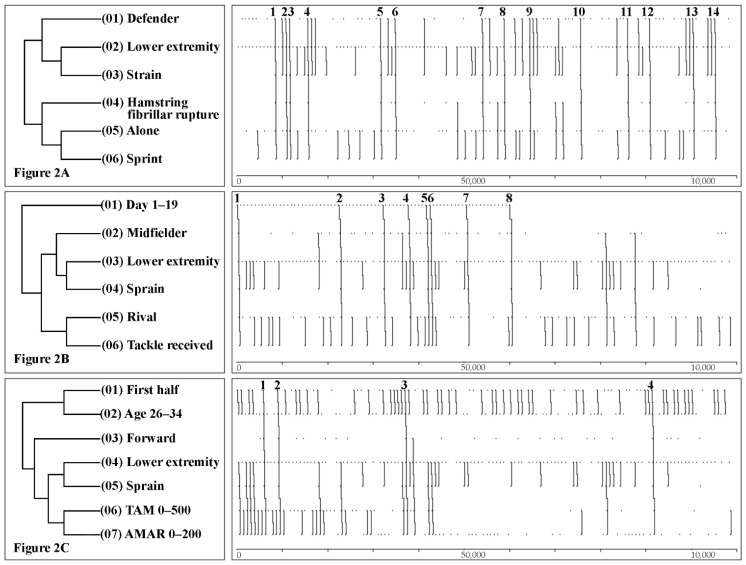
T-patterns of the investigation: (**A**) defender, (**B**) midfielder and (**C**) forward.

**Figure 3 ijerph-19-00252-f003:**
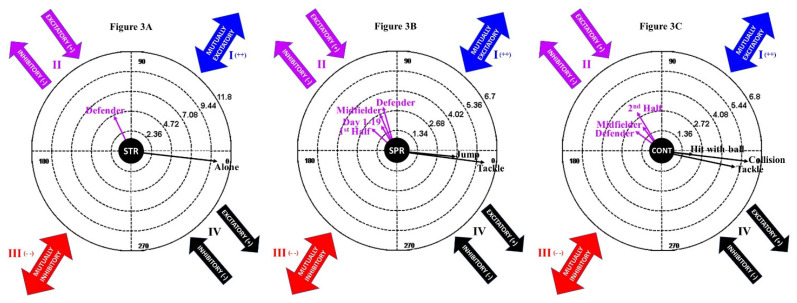
Polar coordinates of the study: (**A**) strain, (**B**) sprain and (**C**) contusion.

**Table 1 ijerph-19-00252-t001:** Descriptive obtained values and χ^2^ test intra-criteria.

Criteria	Category	Subcategory	n	%	χ^2^ (*p*-Value)	Criteria	Category	Subcategory	n	%	χ^2^ (*p*-Value)
Injury	SPRAIN		33	24.3	5.327 (0.070) ^a^	Time	1st Half: 0′–45′		74	54.4	1.441 (0.230)
	Ankle sprain		15	11.0			0′–15′		25	18.4	
	Knee sprain		8	5.9			16′–30′		27	19.9	
	Acromioclavicular sprain		1	0.7			31′–45′ + added time		22	16.2	
	Anterior cruciate ligament rupture		9	6.6			2nd Half: 46′–90′		62	45.6	
	STRAIN		45	33.1			46′–60′		31	22.8	
	Hamstring fibrillar rupture		25	18.4			61′–75′		17	12.5	
	Quadricep fibrillar rupture		5	3.7			76′–90′ + added time		14	10.3	
	Soleus-gastrocnemius fibrillar rupture		3	2.2		How the injury took place	ALONE		71	52.2	0.265 (0.607) ^c^
	Adductor fibrillar rupture		11	8.1			Sprint		33	24.3	
	Psoas fibrillar rupture		1	0.7			Turn		8	5.9	
	CONTUSION		26	19.1			Shooting		9	6.6	
	Head, face or neck contusion		3	2.2			Ball control		2	1.5	
	Lower extremity contusion		20	14.7			RIVAL		65	47.8	
	Trunk contusion		3	2.2			Jump		19	14.0	
	FRACTURE		8	5.9			Collision		13	9.6	
	Head, face or neck fracture		3	2.2			Received tackle		37	27.2	
	Trunk fracture		1	0.7			Performed tackle		12	8.8	
	Upper extremity fracture		2	1.5			Hit by ball		2	1.5	
	Lower extremity fracture		2	1.5			Goalkeeper′s save		1	0.7	
	DISLOCATION		4	2.9		Player laterality	Right-footed		104	76.5	39.474 (0.000) ^b^
	Upper extremity dislocation		3	2.2			Left-footed		31	22.8	
	Lower extremity dislocation		1	0.7			Ambidextrous		1	0.7	
	OVERUSE		14	10.3		Age	<18 years		0	0.0	77.882 (0.000) ^b^
	Hamstring overuse		6	4.4			18–25 years		44	32.4	
	Gluteus overuse		1	0.7			26–34 years		88	64.7	
	Adductor overuse		3	2.2			>34 years		4	2.9	
	Quadriceps overuse		3	2.2		Position	Goalkeeper		7	5.1	21.767 (0.000) ^b^
	Gastrocnemius overuse		1	0.7			Defender		67	49.3	
	OTHERS (wound, concussion, etc.)		6	4.4			Midfielder		37	27.2	
Months	August		8	5.9	33.559 (0.000)		Forward		25	18.4	
	September		22	16.2		Total accumulated minutes	0′–500′		46	33.8	9.588 (0.022)
	October		18	13.2			501′–1000′		39	28.7	
	November		9	6.6			1001′–1500′		23	16.9	
	December		7	5.1			>1500′		28	20.6	
	January		15	11.0		Accumulated minutes after resting (>7 days)	0′–200′		68	50.0	60.294 (0.000)
	February		15	11.0			201′ –400′		41	30.1	
	March		10	7.4			401′–600′		15	11.0	
	April		27	19.9			>600′		12	8.8	
	May		5	3.7		Zone	DEFENSIVE ZONE		93	68.4	18.382 (0.000)
Stadium	Local		67	49.3	0.029 (0.864)		*Zone 1*		21	15.4	
	Visitor		69	50.7			*Zone 2*		24	17.6	
Injury location	Head, face or neck		8	5.9	276.882 (0.000)		*Zone 3*		24	17.6	
	Lower extremity: from waist to feet		118	86.8			*Zone 4*		24	17.6	
	Upper extremity: from shoulder to hands		5	3.7			OFFENSIVE ZONE		43	31.6	
	Trunk-back: from neck to waist		5	3.7			*Zone 5*		12	8.8	
Leg injury	Right leg injuries		64	47.1	0.847 (0.357) ^b^		*Zone 6*		16	11.8	
	Left leg injuries		54	39.7			*Zone 7*		7	5.1	
	No leg injury		18	13.2			*Zone 8*		8	5.9	
Moment in the season	Days 1–19		76	55.9	1.882 (0.170)						
	Days 20–38		60	44.1							

Note. ^a^ The chi-square test was performed among the three categories with the highest frequency; ^b^ the chi-square test was calculated by eliminating the lowest-frequency category; ^c^ the chi-square test was calculated with the categories (alone-rival).

**Table 2 ijerph-19-00252-t002:** Analysis of the selective search for T-patterns.

Description of T-Patterns with Three Occurrences	N
Total T-patterns detected	8571
Non-useful T-patterns (do not meet selection criterion)	7773 (91%)
T-patterns not excluded	798 (9%)
T-patterns with strain, sprain or contusion	2461
T-patterns with defender, midfielder or forward	3170
T-patterns with defender and strain	450
T-patterns with defender and sprain	55
T-patterns with defender and contusion	63
T-patterns with midfielder and strain	11
T-patterns with midfielder and sprain	121
T-patterns with midfielder and contusion	46
T-patterns with forward and strain	21
T-patterns with forward and sprain	26
T-patterns with forward and contusion	5

**Table 3 ijerph-19-00252-t003:** T-patterns according to position (defender, midfielder and forward) and most frequent injury (strain, sprain and contusion).

T-Pattern	O	L
((defender (lower extremity strain)) (hamstring fibrillar rupture (alone sprint)))	14	6
((defender (lower extremity strain)) AMAR 0–200)	18	4
(defensive zone ((defender (lower extremity sprain)) (ankle sprain (alone jump))))	4	7
(defender ((lower extremity contusion) (lower extremity contusion rival)))	8	5
(day 20–38 (defensive zone (defender ((lower extremity contusion) (lower extremity contusion rival))))	5	7
((midfielder (lower extremity strain)) (hamstring fibrillar rupture alone))	3	5
((day 1–19 first half) ((age 26–34 midfielder) (lower extremity strain)))	3	5
(v1 ((mid (lower extremity sprain))(rival tackle received)))	8	6
((midfielder (lower extremity sprain)) TAM 0–500)	8	4
((day 1–19 second half) (defensive zone (midfielder contusion)))	5	5
(midfielder (contusion rival))	8	3
(((day 1–19 first half) (offensive zone ((forward lower extremity) (strain TAM 0–500)))) AMAR 200–400)	3	8
offensive zone ((forward (lower extremity strain)) (alone sprint)))	6	6
((first half age 26–34) (forward ((lower extremity sprain) (TAM 0–500 AMAR 0–200))))	4	7
(forward sprain) (rival tackle received))	4	4
(forward (contusion rival))	3	3
(second half (forward contusion))	3	3

Note. TAM: total accumulated minutes; AMAR: accumulated minutes after resting. O: occurrence; L: length.

## Data Availability

Not applicable.

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
