# Peer review of "Patterns of Injury in the Spanish Football League Players"

_ijerph, 2021, doi:10.3390/ijerph19010252_

Round 1

Reviewer 1 Report

Congratulations to the authors about high quality of the manuscript entitled: Injury patterns in the Spanish Football League. The authors point that the problem of most of the studies is that they are especially descriptive, without establishing what causes the injuries. That is way, the objective of this study is to discover and analyze the injury patterns of the Spanish football league during the season 2016/17. The results of this research generate knowledge of the injury patterns of the football players that will be useful to the coaches, trainers, and players to improve the training and competition systems

  • My first analysis is about the research title, the word “players” can be inserted in title. Because the people are those suffer with these problems. Example: Injury patterns in the Spanish players from the Football League
  • It is related to methods suggestion in Participants section, line 77, as follow: In any research field, we don’t to refer about participants as “injuries”, it is the research object. The participants are people, in this case, the football players. Considering this, the players that suffer injuries should be consulted about your participating in the study. How the authors invite the injuries to do the present study? How the authors have authorization to research came from the injuries? The injuries sign the free consent term? It is a problem to be solved in my point of view about ethic in science.
  • About the Table 2. Descriptive obtained values and the link between the categories. LINE 155. There is a exacerbate number of abbreviations in this table. It is impossible to reader understand the table, even that reading being linked with text. The same characteristic happens in another images and representations.

Reviewer 2 Report

Please, review the English.

Also, try to improve the practical applications and the discussion section trying to give a more precise conclusions.

Reviewer 3 Report

Thank you for submitting your manuscript to this quality journal and sorry for the delay of my review. After carefully reviewing your manuscript, the followings are my suggestions for quality improvement:

  1. In your introductory part, it would be better to include more in-depth discussion on “Spanish football league and its sport.” In that way, potential audience can comprehend the circumstance of your league and also most frequent injury types.
  2. On line 73, you mentioned “other authors” I believe you are indicating other relevant theories of the observational design. If it is true, you should have your manuscript reviewed by English editing service or colleague who are more familiar with that kind of expression.
  3. In line 78, de 380 matches….must be typing error. Needs to be fixed.
  4. For the 2.2.(Participants), if possible, you should provide more detailed information on potential participants of your study and also about study circumstances such as why and how those injuries may occur
  5. The weakest part of you manuscript is “instruments” – in order to provide reliable findings from your study, you should provide more detailed description on your observation instrument in terms of “selection criterion”, and unique aspects of your instruments. Could it be reliable enough to be utilized for sports participants and their injuries?
  6. In line 95, other authors again – please change it properly
  7. In line 120, instead of that statistical significance was taken upon p<0.05. This statement doesn’t sound right. I understand what you mean but. Please rewrite your data analysis parts using right terminologies (i.e., assessing big difference – not right expression).  

Round 2

Reviewer 3 Report

Authors successfully addressed those issues upon my requests. 

It is now ready to be published